# Extent of Spin Contamination Errors in DFT/Plane-wave Calculation of Surfaces: A Case of Au Atom Aggregation on a MgO Surface

**DOI:** 10.3390/molecules24030505

**Published:** 2019-01-30

**Authors:** Kohei Tada, Tomohiro Maruyama, Hiroaki Koga, Mitsutaka Okumura, Shingo Tanaka

**Affiliations:** 1Research Institute of Electrochemical Energy, National Institute of Advanced Industrial Science and Technology (AIST), 1-8-31, Midorigaoka, Ikeda, Osaka 563-8577, Japan; swing-tanaka@aist.go.jp; 2Department of Chemistry, Graduate School of Science, Osaka University, 1-1, Machikaneyama, Toyonaka, Osaka 560-0043, Japan; maruyamat16@chem.sci.osaka-u.ac.jp (T.M.); ok@chem.sci.osaka-u.ac.jp (M.O.); 3Elements Strategy Initiative for Catalysts and Batteries (ESICB), Kyoto University, 1-30 Goryo Ohara, Nishikyo, Kyoto 615-8245, Japan; koga.hiroaki.6u@kyoto-u.ac.jp

**Keywords:** spin contamination error, open-shell structure, static electronic correlation, Au cluster, gold catalyst, dimerization, molecule/surface interaction

## Abstract

The aggregation of Au atoms onto a Au dimer (Au_2_) on a MgO (001) surface was calculated by restricted (spin-un-polarized) and unrestricted (spin-polarized) density functional theory calculations with a plane-wave basis and the approximate spin projection (AP) method. The unrestricted calculations included spin contamination errors of 0.0–0.1 eV, and the errors were removed using the AP method. The potential energy curves for the aggregation reaction estimated by the restricted and unrestricted calculations were different owing to the estimation of the open-shell structure by the unrestricted calculations. These results show the importance of the open-shell structure and correction of the spin contamination error for the calculation of small-cluster-aggregations and molecule dimerization on surfaces.

## 1. Introduction

Dimerization and its reverse process, homolytic cleavage, are basic chemical reactions, which are important for investigating heterogeneous reactions on surfaces [1,2,3,4,5,6,7,8,9,10,11,12]. For instance, dissociation of hydrogen on metal surfaces is an elementary reaction for hydrogenation by heterogeneous metal catalysts, and dimerization of nitrogen atoms and nitric monoxide (NO) molecules are regarded as elemental reactions for NO_x_ elimination from exhaust gases. In addition, both dissociation and dimerization of metal clusters on their support surfaces are crucial for clarifying the degradation mechanism of nanosized metal catalysts, especially gold catalysts [10,13,14,15,16,17,18].

We cannot estimate the potential energy curves for the dimerization of atoms (or/and radical molecules) and hemolytic cleavage of molecules by first-principle calculations with spin-unpolarized (spin-restricted) single reference schemes [19]. When the monomer has unpaired electrons, the reaction systems have an open-shell structure; however, spin-unpolarized calculations cannot estimate the open-shell structure, and the estimated values include artificial unstabilization. The artificial error is called static correlation. To avoiding this error, we perform multi-reference calculations or spin-polarized (spin-unrestricted) calculations [19,20,21,22,23,24,25,26,27].

Thus, multi-reference or spin-polarized schemes are adopted for the first-principle calculations of the potential energy curves for dimerization. The multi-reference calculations are generally more accurate than spin-polarized calculations because the methods account for the multifigurational states, but the computational costs are very high [20,21]. In contrast, the computational costs of spin-polarized calculations are not very high, so these methods can be applied to surface reaction systems, but the methods include an artificial error, such as spin contamination error [22,23,24,25,26,27,28,29,30,31,32,33,34,35,36]. However, it was shown that relevant corrections result in the same potential energy curves as those by multireference calculations even when spin-polarized calculations are used [31,32,33,34].

For first-principle calculations of heterogeneous catalysts, large slab surface models, large metal cluster models, or both are needed. The models consist of hundreds to thousands of atoms, and multireference calculations cannot be adopted due to the high computational costs. Therefore, we adopt the spin-polarized single reference scheme for calculating the adsorptions of molecules that have open-shell structures. Generally, the calculations are based on the density functional theory (DFT) [37], the core electrons are treated by pseudo potentials [38,39,40], and the wave functions are expanded by plane-wave basis sets (DFT/plane-wave method).

The effects of spin contamination error and static correlation on dimerizations (homolytic cleavages) in the gas phase are well known, and the errors can be corrected. By contrast, the errors that are included in DFT/plane-wave calculations for dimerizations (hemolytic cleavages) on surfaces have not been corrected, even though it is possible that the errors affect the results of DFT/plane-wave calculations. This is due to the difficulty of the corrections.

Calculating dimerizations on a surface by DFT/plane-wave, large supercells are necessary for ignoring artificial interactions between a given adsorbate and the adsorbate in the neighboring cell. Hence, it was difficult to perform even spin-polarized calculations owing to the high computational costs. However, with the availability of high-performance of computers, high-cost spin-polarized DFT/plane-wave calculations have become feasible; i.e., nowadays, we can estimate the effects of static correlation on the DFT/plane-wave calculations for surface reactions.

On the other hand, there was no scheme that can estimate the errors in DFT/plane-wave calculations, for correcting spin contamination errors. However, it was shown that the approximate spin projection (AP) scheme [28,41] for the correction of spin contamination errors can be applied to the DFT/plane-wave method (AP-DFT/plane-wave) [35,36]. By using AP-DFT/plane-wave method, we can investigate the effects of spin contamination on surface reactions. Recently, it was clarified that (1) the spin contamination error affects the interaction energy between Au atoms dissociating on a MgO (001) surface [35], (2) the rate-determining step for NO dimerization on a ZrO_2_/Cu catalyst model is changed when the error is corrected [36], and (3) the error occurs in NO dimerization on a TiO_2_/Ag catalyst model even when the NO-NO distance is outside the region where the spin contamination error does not occur in the gas phase [36].

Thus, it has become possible to investigate the effects of spin contamination and static correlation on DFT/plane-wave calculations for describing the potential energy surfaces of surface reactions, although detailed investigations are yet to be performed. The present study is the first to investigate the aforementioned effects. The investigated surface reaction is the aggregation of Au atoms on a MgO (001) surface (Au_2_ dissociation on the MgO (001) surface) shown in Equation (1):(1)$$(\mathrm{Au}+\mathrm{Au})/\mathrm{MgO}⇌{\mathrm{Au}}_{2}/\mathrm{MgO}$$

The Au/MgO system is a typical gold catalyst system [14,16,35,42,43,44], and reaction (1) is related to the sintering of Au clusters on the MgO surface, which pertains to a crucial degradation of the gold catalyst [13,14,15,16,17,18], and its suppression. The potential energy curves of the closed singlet (LS: low spin), triplet (HS: high spin), and open singlet (BS: broken symmetry, which is a common name of open-shell structures estimated by spin-polarized calculations) states were estimated, and the spin contamination error in the open singlet state was estimated by the AP scheme (AP: BS state with AP correction). In addition, the corresponding gas-phase reaction (Equation (2)) was calculated in the same manner as the surface reaction (Equation (1)):(2)$$\mathrm{Au}+\mathrm{Au}⇌{\mathrm{Au}}_{2}$$

Using the calculation results, we investigated the effects of the errors on reaction (1) and the effects of molecule/surface interactions on the magnitude of the errors.

## 2. Computational Procedure

### 2.1. Method

The wave functions were calculated by the DFT scheme [37] with the PW91 exchange-correlation functional [45]. Spin-polarized DFT was adopted for calculating the open-shell structures (HS and BS), and spin-unpolarized DFT was adopted for the closed-shell structure (LS). For the interpolation of the correlation part, the formula proposed by Vosko, Wilk, and Nusair was used [46]. The core electrons were treated by the ultra-soft-pseudo-potential method [38], and the numbers of valence electrons of Mg, O, and Au were 2, 6, and 11, respectively. The energy cut-off values were 400 eV (wave function) and 2400 eV (augmented charge). The only Γ-point was calculated; the energy dependence on *k*-point sampling has already been investigated in our prior work [35]. The DFT calculations were performed by the Vienna Ab initio Simulation Package (VASP) [47,48,49,50]. For population analysis, the Bader charge analysis [51,52,53,54] was used. The Visualization for Electronic and STructural Analysis program (VESTA) [55] was used for visualization of the geometry and spin density distribution.

### 2.2. AP Scheme

The spin contamination errors in the energies of the open singlet states were estimated by the AP scheme [28,41]. In the AP method, the spin contamination error is corrected via a projection to the Heisenberg Hamiltonian:(3)$${\hat{H}}_{\mathrm{HB}}=-2J_{\mathrm{ab}}{\hat{S}}_{a}{\hat{S}}_{b}$$
${\hat{S}}_{x}$
is a spin operator on site x, and $J_{\mathrm{ab}}$ is the effective magnetic exchange interaction between spins. The projection on the Heisenberg Hamiltonian indicates that Equations (4) and (5) are established:(4)$$J_{\mathrm{ab}}=\frac{E_{\mathrm{HB}}^{\mathrm{LS}}-E_{\mathrm{HB}}^{\mathrm{HS}}}{{\langle {\hat{S}}^{2}\rangle }_{\mathrm{HB}}^{\mathrm{HS}}-{\langle {\hat{S}}^{2}\rangle }_{\mathrm{HB}}^{\mathrm{LS}}}=\frac{E_{\mathrm{SA}}^{\mathrm{LS}}-E_{\mathrm{SA}}^{\mathrm{HS}}}{{\langle {\hat{S}}^{2}\rangle }_{\mathrm{exact}}^{\mathrm{HS}}-{\langle {\hat{S}}^{2}\rangle }_{\mathrm{exact}}^{\mathrm{LS}}}$$
(5)$$J_{\mathrm{ab}}=\frac{E_{\mathrm{HB}}^{\mathrm{LS}}-E_{\mathrm{HB}}^{\mathrm{HS}}}{{\langle {\hat{S}}^{2}\rangle }_{\mathrm{HB}}^{\mathrm{HS}}-{\langle {\hat{S}}^{2}\rangle }_{\mathrm{HB}}^{\mathrm{LS}}}=\frac{E_{U}^{\mathrm{LS}}-E_{U}^{\mathrm{HS}}}{{\langle {\hat{S}}^{2}\rangle }_{U}^{\mathrm{HS}}-{\langle {\hat{S}}^{2}\rangle }_{U}^{\mathrm{LS}}}$$

Here, *E* and $\langle {\hat{S}}^{2}\rangle$ are the expectation values of the Hamiltonian $\langle \Psi |\hat{H}|\Psi \rangle$ and the square of the total spin operator $\langle \Psi |{\hat{S}}^{2}|\Psi \rangle$, respectively. The superscripts LS and HS indicate the values of the low spin state and high spin state, respectively. The subscripts (HB, SA, exact, and U) show the method for estimating the expectation values; namely, HB is the value of the Heisenberg model Hamiltonian, SA is estimated by highly accurate spin-adapted calculations such as multireference calculations, exact means the exact value (the eigenvalue of ${\hat{S}}^{2}$), and U indicates the value estimated by spin-polarized (unrestricted) calculations.

Because the spin contamination error is artificial contamination from the higher spin state [19], the HS state is not affected by the error. Therefore, the HS values of U and SA are approximately the same, and we obtain:(6)$$E_{\mathrm{SA}}^{\mathrm{LS}}=\frac{{\langle {\hat{S}}^{2}\rangle }_{\mathrm{exact}}^{\mathrm{HS}}-{\langle {\hat{S}}^{2}\rangle }_{\mathrm{exact}}^{\mathrm{LS}}}{{\langle {\hat{S}}^{2}\rangle }_{\mathrm{exact}}^{\mathrm{HS}}-{\langle {\hat{S}}^{2}\rangle }_{U}^{\mathrm{LS}}}E_{U}^{\mathrm{LS}}+\frac{{\langle {\hat{S}}^{2}\rangle }_{\mathrm{exact}}^{\mathrm{HS}}-{\langle {\hat{S}}^{2}\rangle }_{U}^{\mathrm{LS}}}{{\langle {\hat{S}}^{2}\rangle }_{\mathrm{exact}}^{\mathrm{HS}}-{\langle {\hat{S}}^{2}\rangle }_{U}^{\mathrm{LS}}}E_{U}^{\mathrm{HS}}$$

The spin contamination error in energy is the difference between $E_{\mathrm{SA}}^{\mathrm{LS}}$ and $E_{U}^{\mathrm{LS}}$:(7)$$\mathrm{SCE}=\frac{{\langle {\hat{S}}^{2}\rangle }_{\mathrm{exact}}^{\mathrm{LS}}-{\langle {\hat{S}}^{2}\rangle }_{U}^{\mathrm{BS}}}{{\langle {\hat{S}}^{2}\rangle }_{\mathrm{exact}}^{\mathrm{HS}}-{\langle {\hat{S}}^{2}\rangle }_{U}^{\mathrm{BS}}}(E_{U}^{\mathrm{HS}}-E_{U}^{\mathrm{BS}})$$

Here, LS is rewritten as BS, because the low-spin state including the spin contamination error has a broken-symmetry electronic structure. The ${\langle {\hat{S}}^{2}\rangle }_{U}$ values were estimated by Wang’s scheme [56]:

### 2.3. Model

A 4 × 4 MgO (001) slab, as shown in Figure 1a, was used for the MgO (001) surface. The lattice constant was optimized by using the MgO bulk structure. The number of atomic layers in the surface model was 4, and the bottom layer was fixed to mimic the MgO bulk structure. The dependencies of the Au atom adsorption energy and O atom desorption energy on the number of atomic layers were checked. Figure 1b shows a schematic view of the calculated reaction, such as aggregation of Au atoms (Au_2_ dissociation) on the MgO (001) surface. The diffusion direction of the Au atom is [010], and the position of the diffusing Au atom (^2^Au) is represented by fractional coordinate r. For the structure shown on the left side of Figure 1b, r = 0.500, and for that on the right side, r = 0.000. By varying r from 0.500 to 0.000 in 0.025 increments, the potential energy of reaction (1) was estimated. The geometry of the MgO slab (except for the bottom layer) and the [001] coordination of the Au atoms were optimized at each r value. Figure 1c–f show the optimized typical structures and their r values.

### 2.4. Definitions of Energies

The energy baseline of this work is the non-adsorbed state, such as Au + Au + MgO. When using this baseline definition, the energy of the calculated ^1^Au^2^Au/MgO system is the same as the adsorption energy shown in Equation (8):(8)Eads=E(A1uA2u/MgO)−E(MgO)−2E(Au)
*E*(^1^Au^2^Au/MgO) is the total energy of the Au_2_ (or ^1^Au and ^2^Au atoms)-adsorbed MgO model, *E*(MgO) is the total energy of the optimized 4 × 4 MgO (001) slab, and *E*(Au) is the total energy of the Au atom in the 30 × 30 × 30 Å supercell that is sufficient volume to calculate Au atom energy. By calculating *E*_ads_ at each r value, the potential energy is estimated.

*E*_ads_ can be divided into four energies; *E*_int_(^1^Au/^2^Au) is the interaction energy between ^1^Au (non-diffused Au atom) and ^2^Au (diffused Au atom) defined by Equation (9); *E*_int_(^1^Au/MgO) is the interaction energy between ^1^Au and MgO defined by Equation (10); *E*_int_(^2^Au/MgO) is the interaction energy between ^2^Au and MgO defined by Equation (11); and *E*_dis_(MgO) is the distortion energy of MgO due to the adsorption of Au atoms defined by Equation (12):(9)Eint(A1u/A2u)=E(A1uA2u/MgO)−E(A1u/MgO)fix−E(A2u/MgO)fix+E(MgO)fix
(10)Eint(A1u/MgO)=E(A1u/MgO)fix−E(MgO)fix−E(Au)
(11)Eint(A2u/MgO)=E(A2u/MgO)fix−E(MgO)fix−E(Au)
(12)$$E_{\mathrm{dis}}\left(\mathrm{MgO}\right)=E{\left(\mathrm{MgO}\right)}_{\mathrm{fix}}-E(\mathrm{MgO})$$
*E*(X) is the total energy of X, and the subscript fix in Equations (9)–(12) indicates that the total energy is calculated using the fixed structure which is in the optimized ^1^Au^2^Au/MgO system.

The previous work [35] argued that the percentage of spin contamination error in *E*_int_(^1^Au/^2^Au) is large. Therefore, the divided energies (*E*_int_(^1^Au/^2^Au), *E*_int_(^1^Au/MgO), *E*_int_(^2^Au/MgO), and *E*_dis_(MgO)) were also investigated. In addition, variations in the effective exchange integral *J*_ab_ (Equation (5)) and the energy difference BS and HS, Δ*E* (Equation (13)), were investigated; these values reflect the stability of the BS state:(13)ΔE=|E(HS)−E(LS)|

## 3. Results and Discussion

All the optimized structures are included in the Appendix A. Figure 2 shows the distance data for each r. *d*(^1^Au-^2^Au) is the distance between ^1^Au and ^2^Au, *d*(^1^Au-^1^O) is the distance between ^1^Au and ^1^O, and *d*(^2^Au-n.n.) is the distance between ^2^Au and its nearest neighbor atom. The nearest neighbor atoms are summarized in the Appendix A.

Figure 3 depicts the calculated potential energy curves for reaction (1); the horizontal axis in Figure 3a,b is r and *d*(^1^Au-^2^Au), respectively. The calculated *E*_ads_ reflects the total energy; therefore, the stability of the electronic states and adsorption structure can be judged from the value of *E*_ads_.

The dependencies of *E*_int_(^1^Au/^2^Au), *E*_int_(^1^Au/MgO), *E*_int_(^2^Au/MgO), and *E*_dis_(MgO) on r are shown in Figure 4. For *E*_int_(^1^Au/^2^Au), the values of the LS (closed singlet state), HS (triplet state), BS (open singlet state without correction of the spin contamination error), and AP (open singlet state with correction of the spin contamination error by the AP scheme) were estimated. For other energies such as *E*_int_(^1^Au/MgO), *E*_int_(^2^Au/MgO), and *E*_dis_(MgO), only the LS, HS, and BS values were estimated. For estimating these energies, the energy of an open singlet structure, which is a state including the spin contamination error, was not used; therefore, these energies do not include the spin contamination error, and the AP values are identical to the BS value. Figure 5 shows the percentage of the four energies in *E*_ads_.

From the results shown in Figure 2a, it is confirmed that Au atoms aggregates to Au_2_ as r decreases. We classified the calculation results with the nearest neighbor atom of ^2^Au, and made discussions. Section 3.1 shows the results for the interaction of ^2^Au with ^3^O (r = 0.500–0.450, typical structure is r = 0.500 shown in Figure 1c). Section 3.2 shows the results for the interaction of ^2^Au with ^2^Mg (r = 0.425–0.325, the typical structure is r = 0.375 shown in Figure 1d). Section 3.3 shows the results for the interaction of ^2^Au with ^2^O (r = 0.300–0.200, the typical structure is r = 0.250 shown in Figure 1e). Section 3.4 shows the results for the interaction of ^2^Au with ^1^Au or ^1^Mg (r = 0.175–0.000, the typical structure is r = 0.000 shown in Figure 1f).

### 3.1. Dissociated Au_2_ Adsorption onto MgO: r = 0.500–0.450

In this r region, r = 0.500–0.450, the Au atoms interact with separate O^2−^ ions. *d*(^1^Au-^2^Au) is long (Figure 2a), and the *E*_int_(^1^Au/^2^Au) values of the open-shell states (HS and BS) are zero (Figure 4a,b). However, the *E*_int_(^1^Au/^2^Au) values of the closed-shell state (LS) are not zero, and the value for r = 0.500 (*d*(^1^Au-^2^Au) = 8.4984 Å) is 0.40 eV. It is difficult to assume that a repulsion of 0.40 eV will experimentally occur; therefore, it is reasonable to assume that the unstabilization is an artificial error, such as static correlation. The spin-unpolarized DFT calculations cannot estimate the potential energy curves of dissociation of the covalent bonds in the gas phase due to the static correlation, and the artificial error occurs as Au_2_ dissociation in the gas phase [35]. The dominant interaction between the Au atoms in Au_2_ is the covalent interaction between the Au 6s orbitals. The highest occupied molecular orbital (HOMO) is a bonding orbital, and the lowest unoccupied molecular orbital (LUMO) is an anti-bonding orbital. The frontier orbitals do not change when Au_2_ is adsorbed onto metal oxides such as MgO [35,43], Al_2_O_3_ [18,57], and TiO_2_ [58,59,60,61]. Hence, the HOMO and LUMO of Au_2_ can be described by using the 6s orbital of ^1^Au ($\varphi_{1}$) and 6s orbital of ^2^Au ($\varphi_{2}$):(14)$$\varphi_{HOMO}=A\left(\varphi_{1}+\varphi_{2}\right)$$
(15)$$\varphi_{LUMO}=B\left(\varphi_{1}-\varphi_{2}\right)$$

A and B are linear combination constants. Then, the frontier orbitals of the calculated ^1^Au^2^Au/MgO systems can be simplified:(16)$$\varphi_{Au-Au}=C\varphi_{1}+D\varphi_{2}$$

In the gas phase, the two Au atoms are identical: |C| = |D|. In the adsorbate/surface system, the two Au atoms are not identical, and then |C|≠|D|. The spin-unpolarized calculations use the same orbitals for electrons with different spins. Using the scheme, the occupied spin orbitals of Au_2_ systems with the 6s orbitals are:(17)$$\{\begin{array}{c} \chi_{1}=\alpha \varphi_{Au-Au}\\ \chi_{2}=\beta \varphi_{Au-Au} \end{array}$$
and the Slater determinant is:(18)$$\begin{array}{c} \frac{1}{\sqrt{2}}\left|\begin{array}{cc} \chi_{1}\left(r_{1},\omega_{1}\right) & \chi_{2}\left(r_{1},\omega_{1}\right)\\ \chi_{1}\left(r_{2},\omega_{2}\right) & \chi_{2}\left(r_{2},\omega_{2}\right) \end{array}\right|\\ =\frac{\mathrm{CD}}{\sqrt{2}}\left|\begin{array}{cc} \alpha \left(\omega_{1}\right)\varphi_{1}\left(r_{1}\right) & \beta \left(\omega_{1}\right)\varphi_{2}\left(r_{1}\right)\\ \alpha \left(\omega_{2}\right)\varphi_{1}\left(r_{2}\right) & \beta \left(\omega_{2}\right)\varphi_{2}\left(r_{2}\right) \end{array}\right|+\frac{\mathrm{CD}}{\sqrt{2}}\left|\begin{array}{cc} \alpha \left(\omega_{1}\right)\varphi_{2}\left(r_{1}\right) & \beta \left(\omega_{1}\right)\varphi_{1}\left(r_{1}\right)\\ \alpha \left(\omega_{2}\right)\varphi_{2}\left(r_{2}\right) & \beta \left(\omega_{2}\right)\varphi_{1}\left(r_{2}\right) \end{array}\right|\\ \;+\frac{\mathrm{CD}}{\sqrt{2}}\left|\begin{array}{cc} \alpha \left(\omega_{1}\right)\varphi_{1}\left(r_{1}\right) & \beta \left(\omega_{1}\right)\varphi_{1}\left(r_{1}\right)\\ \alpha \left(\omega_{2}\right)\varphi_{1}\left(r_{2}\right) & \beta \left(\omega_{2}\right)\varphi_{1}\left(r_{2}\right) \end{array}\right|+\frac{\mathrm{CD}}{\sqrt{2}}\left|\begin{array}{cc} \alpha \left(\omega_{1}\right)\varphi_{2}\left(r_{1}\right) & \beta \left(\omega_{1}\right)\varphi_{2}\left(r_{1}\right)\\ \alpha \left(\omega_{2}\right)\varphi_{2}\left(r_{2}\right) & \beta \left(\omega_{2}\right)\varphi_{2}\left(r_{2}\right) \end{array}\right| \end{array}$$

Here, $r_{x}$ and $\omega_{x}$ are the geometry coordination and spin coordination of electron x. The third and fourth terms of Equation (18) are ion terms, whose schematic view is shown in Figure 6. These terms cause static correlation because they are artificially included and unstable in dissociated structures. Even if |C|≠|D|, e.g., in an adsorbed system, the terms do not disappear and affect the calculated energy. In other words, even if we consider a surface reaction system, when the dissociated adsorbates have localized spins, the DFT/plane-wave calculations of the aggregation reactions and dissociation reactions include static correlation, and the calculation fails. This is similar to the failure of spin-unpolarized calculations of the dissociations of diatomic molecules composed of two elements. Thus, the calculation results (Figure 3 and Figure 4) explicitly show the importance of investigating the open-shell structure by DFT/plane-wave calculations of the surface reaction.

Figure 7 summarizes the calculation results for the gas-phase reaction, reaction (2). The calculation results are obtained by removing MgO from the optimized ^1^Au^2^Au/MgO system and calculating the fixed structures. The results of *E*_int_(^1^Au/^2^Au) for the surface reaction (Figure 4a) and those for the gas-phase reaction (Figure 7a) clearly differ; the former are constant, while the latter are not constant. The reason for this trend can also be inferred from Equation (18). Because of the electrostatic interactions with the MgO surface, the energy difference between the ion (third and fourth) and radical (first and second) terms fluctuates.

### 3.2. Dissociating Au_2_ Adsorption onto MgO (1): r = 0.425–0.325

In this r region, the dissociating Au (^2^Au) interacts with ^2^Mg. The potential energy surface (Figure 3) shows that the transition state of reaction (1) is included in this r region. The structures in which the ^2^Au atom is adsorbed around the ^2^Mg atom are the least stable. However, the heights of the reaction (1) (aggregation of Au atoms) of LS, HS, BS, and AP are different: 0.17, 0.45, 0.40 and 0.32 eV, respectively (Table 1).

The LS states are the least stable among calculated states. Since the LS state at r = 0.500 (the dissociated structure) is also unstable, the activation barrier of the aggregation reaction is estimated to be low (0.17 eV). Assuming that the energy of the open-shell structure is adopted as the baseline (the energy for the dissociated adsorption structure: r = 0.500), the activation barrier is 0.53 eV. The instability is due to the static correlation. However, the effects on the surface reaction at r = 0.325–0.425 (Figure 4a) are smaller than those of the gas-phase reaction (Figure 7a). This is because the ion terms in Equation (18) for the surface system is not as unstable as those of the gas phase. There are electrostatic interactions such as A1uδ+⋯O12− and A2uδ−⋯M2g2+ in the ^1^Au^2^Au/MgO system. The results of Bader charge analysis (Figure 8a,b) show the charge polarization such as ${\mathrm{Au}}^{\delta +}\cdots {\mathrm{Au}}^{\delta -}$. The Bader charges also show that the LS state overestimates the polarization as compared with the BS state. The overestimation causes a large distortion of the MgO slab (*E*_dis_(MgO)) and the LS state becomes more unstable.

The second least stable state is the HS state. In the HS state, there is no artificial unstabilization at r = 0.500; therefore, the activation barrier of the aggregation reaction is the highest (0.45 eV). This value is higher than that of AP by 0.1 eV, and it is obvious that the HS calculation misestimates the activation barrier of reaction (1). In addition, due to the lack of ${\mathrm{Au}}^{\delta +}\cdots {\mathrm{Au}}^{\delta -}$ polarization, the absolute values of *E*_int_(^1^Au/^2^Au) and *E*_dis_(MgO) are smaller than those for the other states.

The BS state is the ground state in the region r = 0.325–0.425, similar to the other r regions. Two interactions stabilize this state: one is the electrostatic interaction between the Au atoms and MgO (similar to that in the LS state at r = 0.325–0.425) shown in Figure 8, and the other is a spin delocalization to the O sites shown in Figure 9 and Figure 10. When the Au atom interacts with the MgO (001) surface, the spin density on the Au atom is delocalized to the interacted O, and Au-O becomes a spin site [35,62,63]. Figure 9 and Figure 10 show that the spin density on the Au atom delocalizes only to the O sites but not to the Mg site; the adsorption of the Au atoms affects the electronic structure of only the O atoms in MgO (001). Thus, the radical spin still remains and localized at the Au-O sites when the Au atoms are adsorbed; therefore, the spin contamination occurs, and the total energy of the BS states is affected by the error.

At this r region, the spin contamination error should be investigated; the results are shown in Figure 11, with related values such as $\langle {\hat{S}}^{2}\rangle$, *J*, and ΔE. The values of SCE, $\langle {\hat{S}}^{2}\rangle$, and *J* of reactions (1) and (2) are summarized in Table 2.

As shown in Figure 11a and Table 2, the spin contamination errors in this r region are large, and the maximum is 0.084 eV. This result is surprising because the spin contamination error does not occur at the Au-Au distances in the gas phase (Figure 7b and Table 2). The spin contamination error affects the calculated energies of BS states of the surface system even at r = 0.325–0.425 because the two Au atoms are not identical due to the interactions with the MgO surface, and a clear energy difference is generated between HS and BS. This mechanism is confirmed by the results of *J* and ΔE, summarized in Table 2.

Figure 3 shows that the spin contamination error also influences the estimation of the activation barrier and the structure of the transition state. The activation barriers without AP correction are 0.40 eV (aggregation reaction of Au atoms) and 2.34 eV (Au_2_ dissociation reaction), while those with AP correction are 0.32 eV and 2.26 eV, respectively. In the aggregation reaction, the percentage of the error is large (20%). In addition, the transition states of BS and AP are different. The former is the structure at r = 0.375 (the ^2^Au atom is at the on-top site of ^2^Mg), while the latter is the structure at r = 0.400, although the energy difference between the structures at r = 0.375 and 0.400 of AP is small (0.01 eV). Considering the asymmetry of the structure of the surface adsorption system, it is not surprising that the position of the ^2^Au atom in the transition state shifts from the on-top site of ^2^Mg. In fact, even before correction, the energies of BS at r = 0.350 and 0.400 are different, and the energy at r = 0.400 is more unstable.

Note that the correction of the spin contamination error in this study was performed only for energy. The corrections to the first derivative (force) and the second derivative (frequency) of energy have not been performed yet, and the activation energies and transition state structures in this study are roughly estimated. Nevertheless, the results of the present work are sufficient to show the importance of the investigation and correction of artificial errors such as spin contamination and static correlation in spin-polarized DFT/plane-wave calculations for surface reactions, because it is obvious that the errors affect the calculated results of the potential energy curves for the aggregation of Au atoms (Au_2_ dissociation) on the MgO (001) surface.

### 3.3. Dissociating Au_2_ Adsorption onto MgO (2): r = 0.300–0.200

In the r region of this section (r = 0.300–0.200), when the Au-Au bond is being created, the ground state is BS and there is an energy difference between HS and BS [35], as confirmed by the potential energy curves shown in Figure 3. The ^2^Au diffusing on MgO interacts with ^2^O, and there are two spin states (^1^Au-^1^O and ^2^Au-^2^O) in the BS states structures of r = 0.200–0.300 (Figure 9).

At r = 0.200, because the covalent bond between the Au atoms remains, the LS state without the spin is more stable than the HS state with spins at the Au-O sites. The BS state does not have any large localized spin, and the electronic state is almost the same as that of LS state, as shown by the results of the $\langle {\hat{S}}^{2}\rangle$ value (Figure 11a), spin density distribution (Figure 9m), and local magnetic moment (Figure 10).

From r = 0.225, the results of the LS, HS, and BS states clearly differ. At r = 0.225–0.300, the ground state is the BS state, and the HS state artificially affects the BS state (spin contamination). The maximum and minimum SCE values of the surface reaction are 0.092 eV (r = 0.225) and 0.009 eV (r = 0.300), respectively. On the other hand, the maximum, minimum, and second minimum SCE values of the gas-phase reaction are 0.0095 eV (r = 0.250), 0.000 eV (r = 0.225), and 0.023 eV (r = 0.300), respectively.

These results indicate that the maximum SCE value of the gas phase is larger than that of the surface reaction. This can be explained by the mechanism reported in our previous study [35]. In the dissociated Au_2_ on MgO, the covalent interaction between the Au atoms is weakened by the interaction between the adsorbed Au and the adsorbing O^2-^. Then, the open singlet state of the surface system becomes less stable than that of the gas-phase system; the energy difference between the HS and BS states decreases, and the SCE value becomes small. SCE(surf.) is smaller than SCE(gas) at r = 0.250–0.300 because of this destabilization of the BS state by the Au/MgO interaction.

However, at r = 0.225, the value of SCE(surf.) is larger than that of SCE(gas) (SCE(gas) is zero), which is explained by the mechanism reported in our other previous study [36]. The Au-Au interaction is weakened by the Au-MgO interaction, which leads to destabilization of not only the BS state but also the LS state. In the surface system, even when the Au-Au distance is short so that there are covalent interactions in the gas phase, there is no clear covalent interaction between the Au atoms, and the LS state of the surface system becomes less stable than the BS state of the surface reaction system. The destabilization of the LS state is confirmed from the results shown in Figure 3a. Because of this destabilization, the BS state becomes the ground state, the spin is localized at the Au-O sites, as shown by the spin density distribution in Figure 9l, and spin contamination occurs.

From the results described in this Section and Section 3.2, it is concluded that in the surface reaction systems, the region where spin contamination affects the total energies becomes wider than that in the gas-phase reaction systems.

On the other hand, Figure 3a and Figure 4a show that the LS state is the least stable among the calculated states at r = 0.250–0.300 due to the static correlation. The artificial unstabilization shown in equation (18) occurs even in the structures where the Au-Au bond is creating (is breaking) at the surface.

The spin contamination and static correlation affect the calculation energies from r = 0.225. Figure 5 shows that the percentages of the Au/Au interaction (*E*_int_(^1^Au/^2^Au)) and the Au/MgO interaction (*E*_int_(^1^Au/MgO) and *E*_int_(^2^Au/MgO)) change from r = 0.225, i.e., from r = 0.225, the Au/MgO interactions become larger than the Au-Au interaction. This result suggests that the spin contamination and static correlation affect the calculation results when the covalent interaction within the adsorbate weakens and the interaction between the dissociated fragments and the surface becomes dominant. This will be used as a criterion for judging whether the effect of artificial errors on surface reactions should be considered. 

### 3.4. Non-Dissociated Au_2_ Adsorption onto MgO: r = 0.175–0.000

In this r region, the distance between ^1^Au and ^2^Au is so short that the ^1^Au-^2^Au covalent interaction is dominant. When there is a clear covalent interaction between ^1^Au and ^2^Au, the open singlet state (BS state) does not exist and converges to the same electronic structure as the closed singlet state (LS state) [35]. Therefore, in this r region, the results of LS and BS identical, and the spin contamination error does not affect the results.

The *d*(^1^Au-^2^Au) values of HS are longer than those of the other states in the region r = 0.000–0.150. This is because the Au atoms do not interact covalently in the r region, as confirmed by the *E*_int_(^1^Au/^2^Au) results of HS shown in Figure 4a.

The *d*(^1^Au-MgO) values of HS are also longer than those of the other states because of the insignificant interaction between the Au atoms of HS. When a Au cluster is adsorbed onto metal oxides, it acquires a slight positive charge and interacts with an O^2−^ ion electrostatically [16,17,18,35,43,44,58,59,60,61,64,65,66,67]. Furthermore, the charge on the Au cluster consisting of two or more Au atoms is polarized, and the Au atoms closer to the surface are more positively charged [17,44,60,65,67]. In other words, when there are Au-Au covalent interactions, the Au cluster can interact with MgO more strongly than the Au atom. In the HS state, which lacks Au-Au covalent interactions, Au_2_ is not polarized and *d*(^1^Au-MgO) becomes longer, as confirmed by the Bader atomic charges shown in Figure 8.

## 4. Conclusions

In order to clarify the effects of spin contamination error and static correlation, the aggregation reaction of Au atoms (Au_2_ dissociation) on the MgO (001) surface have been investigated in detail. The calculation results show that the singlet state calculated by spin-polarized DFT is the ground state throughout the reaction, and that the calculated energy is affected by the spin contamination error. The investigated reaction can be divided into four steps:(1)Two isolated Au atoms are adsorbed onto the on-top of O site in MgO (001), which is the initial state of the aggregation of Au atoms(2)One Au atom interacts with Mg and the other Au atom interacts with O, and there is a slight electrostatic interaction between the Au atoms, which includes the transition state of the Au atom aggregation(3)Generation of Au-Au covalent interaction(4)Non-dissociated adsorption state of Au_2_ on MgO, which is the final structure of the Au atom aggregation.

The spin-unpolarized calculation that gives only the closed-shell structure (LS) can calculate step (4) accurately, but the calculation fails at other steps due to the static correlation. Using the results of the HS states, we can estimate the energy of step (1). However, the HS state calculation misestimates the Au-Au interaction and cannot treat steps (2–4). The BS calculation results, which are the ground state of all the steps, are influenced by the spin contamination error, except for steps (1) and (4). Therefore, the BS calculations overestimate the activation barrier and underestimate the Au-Au interaction energy; for treating all the steps, AP calculation is needed. These results explicitly show the importance of the open-shell structure calculations (spin-polarized DFT/plane-wave) with a spin contamination error correction for investigating the aggregation of small metal clusters on metal oxide surfaces.

It is shown that “the maximum value of the spin contamination error in surface reactions is smaller than that in gas-phase reactions, whereas the area causing spin contamination in the surface reaction is wider than that in the gas-phase reaction.” In addition, it is predicted that “the effects of spin contamination and static correlation emerge when the dissociating adsorbate/surface interaction becomes larger than the covalent interaction in the adsorbate.” These will serve as a useful guide for correcting the errors in spin-polarized DFT/plane-wave calculations for surface reactions.

The spin contamination in force and frequency has not been corrected. For an accurate estimation of activation energies and transition states, corrections such as the AP-opt scheme [29,30] are indispensable. Developing a new method (AP-opt-DFT/plane-wave) that enables corrections is one of the important issues to be resolved in the future.

## Figures and Tables

**Figure 1 molecules-24-00505-f001:**
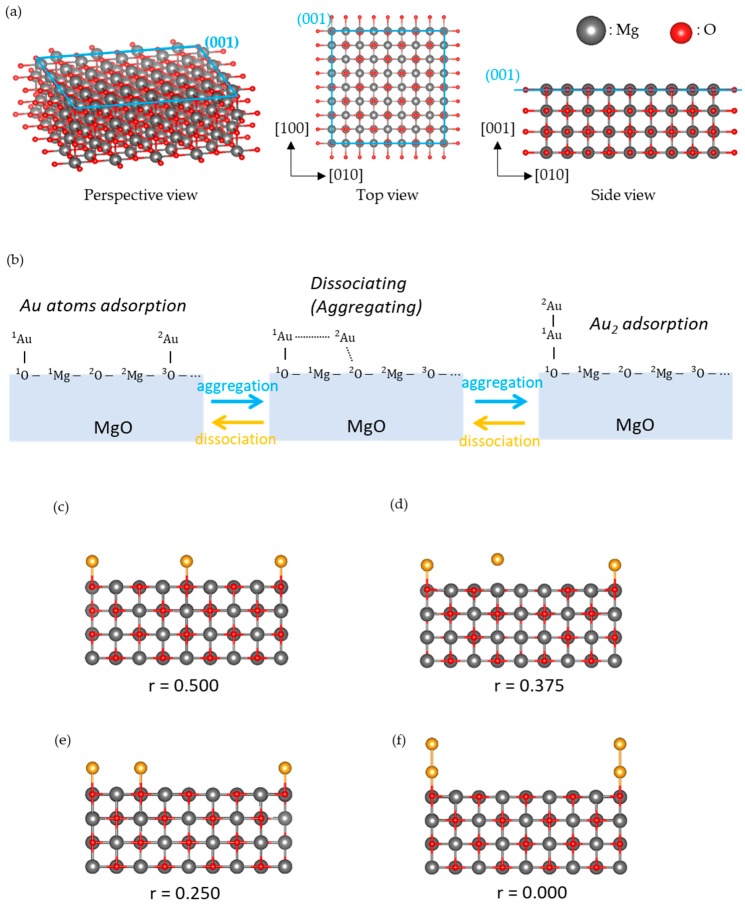
Explanation of calculated models. (**a**) Optimized periodic MgO (001) surface model, (**b**) a schematic view of the calculated surface reaction (Equation (1) in the main text: aggregation of Au atoms or Au_2_ dissociation on MgO (001)). (**c**–**f**) the optimized structures of BS states at r = 0.500 (**c**), r = 0.375 (**d**), r = 0.250 (**e**), and r = 0.000 (**f**); r indicates a fractional coordinate of ^2^Au (diffusing Au atom).

**Figure 2 molecules-24-00505-f002:**
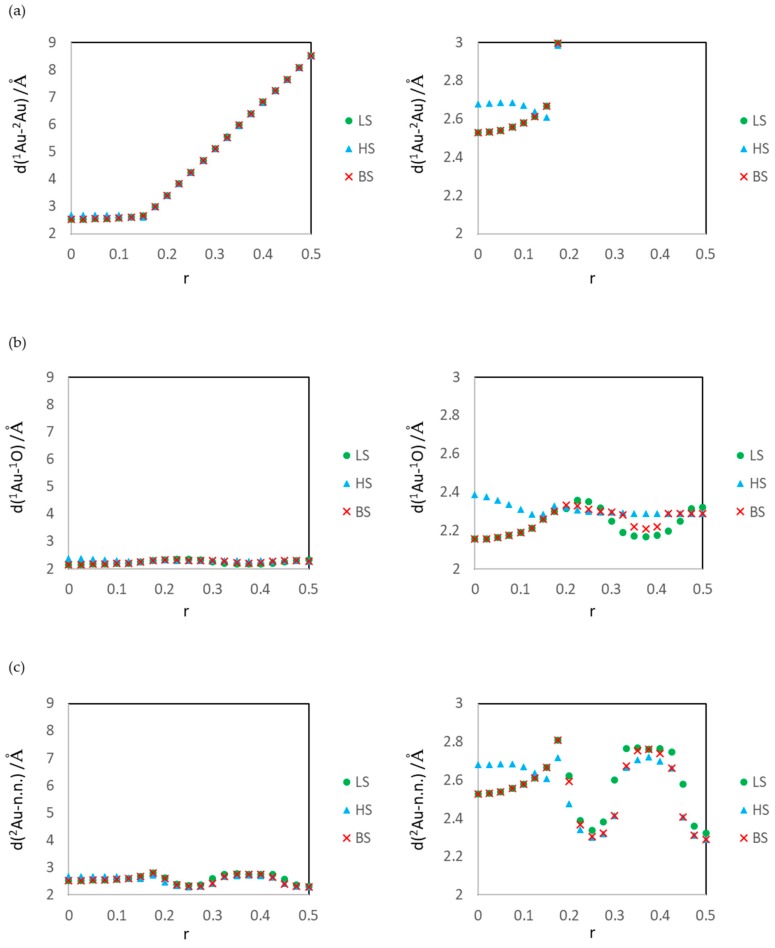
Distance variation of LS, HS, and BS states versus fractional coordinate of ^2^Au: r. (**a**) distance between Au atoms *d*(^1^Au-^2^Au), (**b**) distance between ^1^Au and ^1^O *d*(^1^Au-^1^O), and (**c**) distance between ^2^Au and its nearest neighbor atom. Right panels are enlarged figures of left panels.

**Figure 3 molecules-24-00505-f003:**
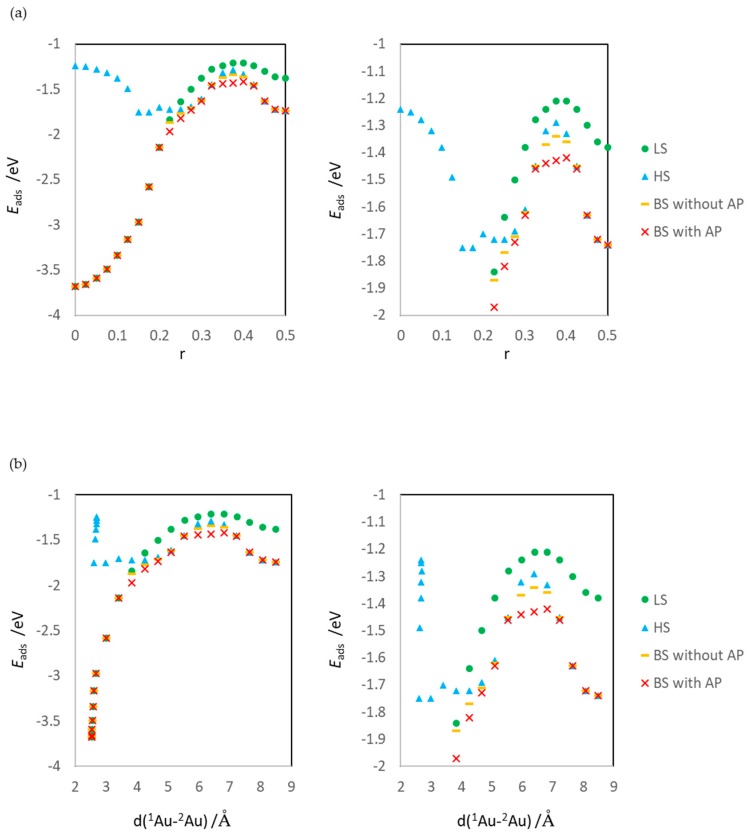
Calculated potential energy curves for reaction (1). (**a**) Energy variation on r and (**b**) on *d*(^1^Au-^2^Au). Right panels are enlarged figures of left panels.

**Figure 4 molecules-24-00505-f004:**
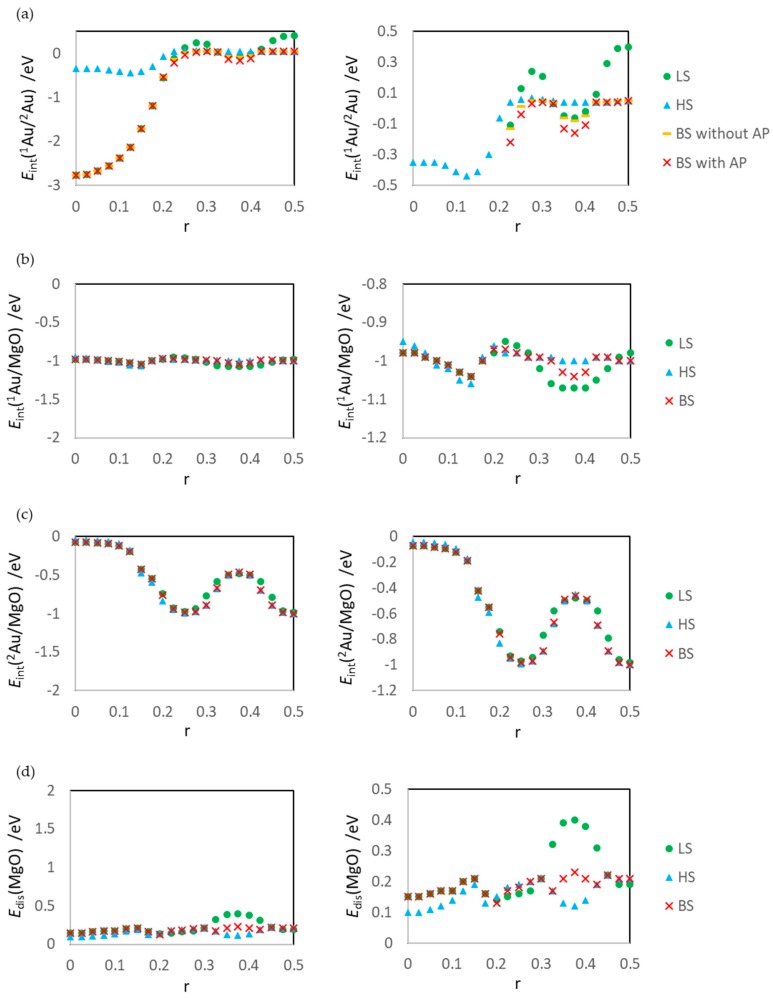
Energy dependence on r. (**a**) Interaction energy between Au atoms *E*_int_(^1^Au/^2^Au) (Equation (9)), (**b**) interaction energy between ^1^Au and MgO surface *E*_int_(^1^Au/MgO) (Equation (10)), (**c**) interaction energy between ^2^Au and MgO surface *E*_int_(^2^Au/MgO) (Equation (11)), and (**d**) distortion energy of MgO surface *E*_dis_(MgO) (Equation (12)). Right panels are enlarged figures of left panels.

**Figure 5 molecules-24-00505-f005:**
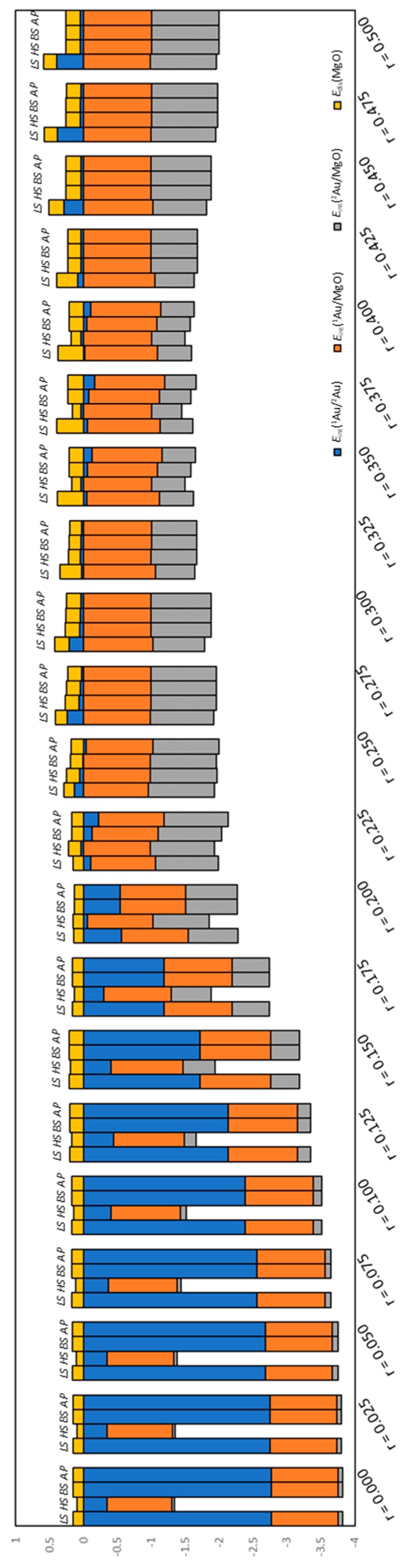
Percentage of *E*_int_(^1^Au/^2^Au), *E*_int_(^1^Au/MgO), *E*_int_(^2^Au/MgO), and *E*_dis_(MgO) in *E*_ads_. The energy unit is eV.

**Figure 6 molecules-24-00505-f006:**
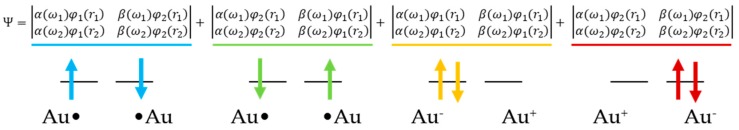
A schematic view of the Slater determinant composed of Au 6s orbitals.

**Figure 7 molecules-24-00505-f007:**
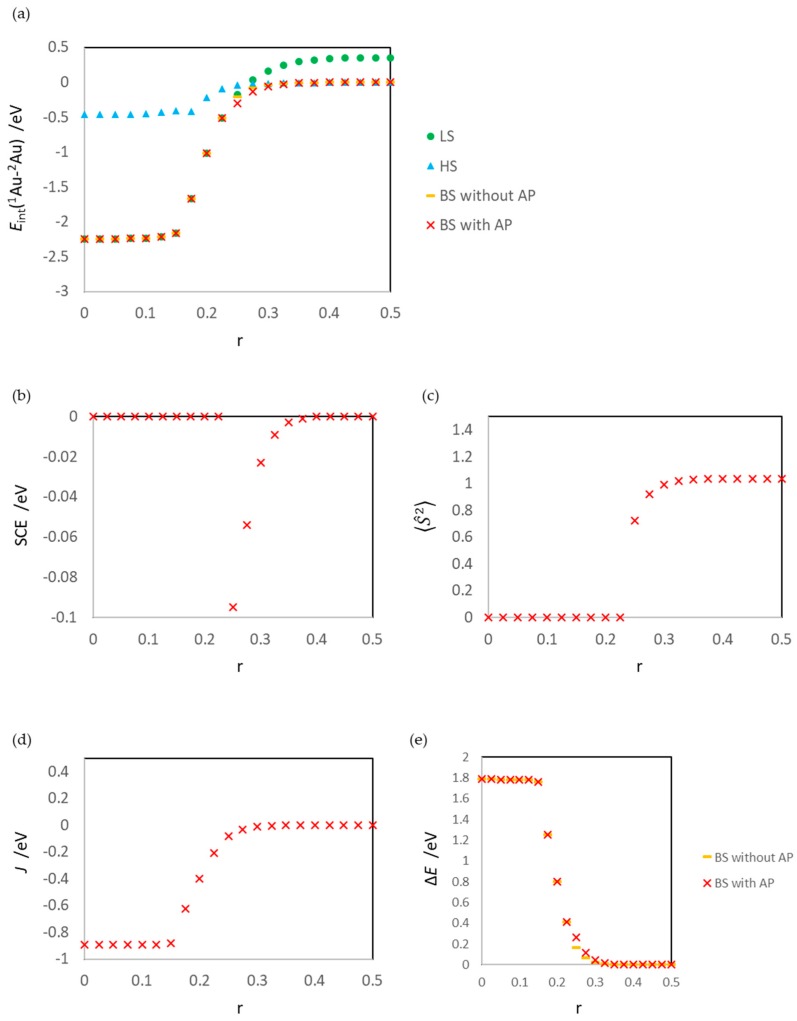
Results of the gas-phase reaction (Au_2_ dissociation), (**a**) interaction energy between Au atoms *E*_int_(^1^Au/^2^Au), (**b**) spin contamination error SCE, (**c**) expectation value of the square of the total spin operator $\langle {\hat{S}}^{2}\rangle$, (**d**) effective exchange integral *J* (Equation (5)), and (**e**) energy difference between the HS and BS states ΔE (Equation (13)). The *J* values for r = 0.000–0.225 are estimated by using LS values; strictly, these values are not effective exchange integrals.

**Figure 8 molecules-24-00505-f008:**
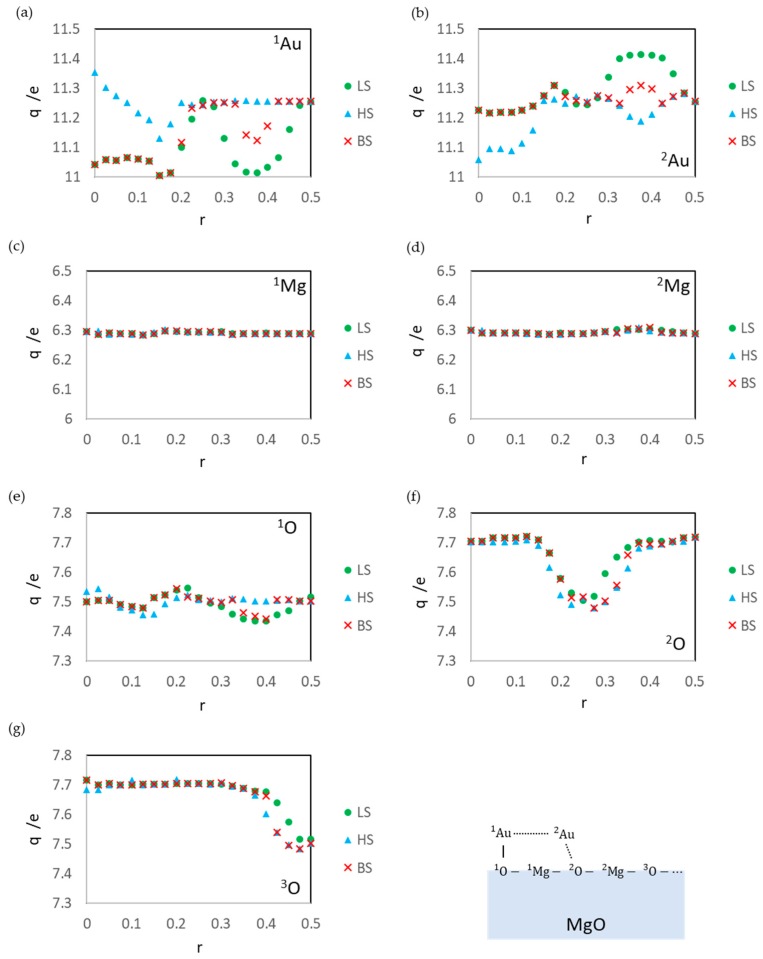
Bader atomic charges q on ^1^Au (**a**), ^2^Au (**b**), ^1^Mg (**c**), ^2^Mg (**d**), ^1^O (**e**), ^2^O (**f**), and ^3^O (**g**) in the optimized structures at each r value.

**Figure 9 molecules-24-00505-f009:**
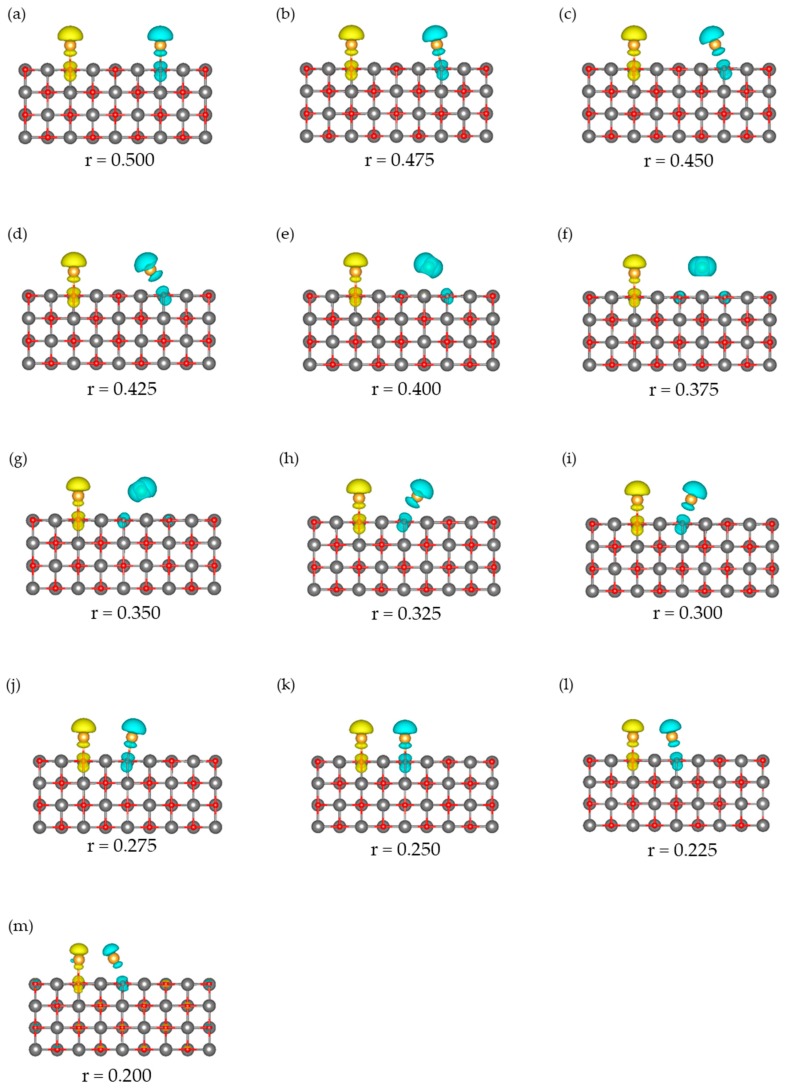
Spin density distributions of BS states. Isosurface values are 0.004 e/Bohr^3^ except for (m). The value of (m) is 1.0 × 10^−9^ e/Bohr^3^. (**a**) r = 0.500, (**b**) r = 0.475, (**c**) r = 0.450, (**d**) r = 0.425, (**e**) r = 0.400, (**f**) r = 0.375, (**g**) r = 0.350, (**h**) r = 0.325, (**i**) r = 0.300, (**j**) r = 0.275, (**k**) r = 0.250, (**l**) r = 0.225, and (**m**) r = 0.200. The spin density distributions of HS states are provided in the Appendix A.

**Figure 10 molecules-24-00505-f010:**
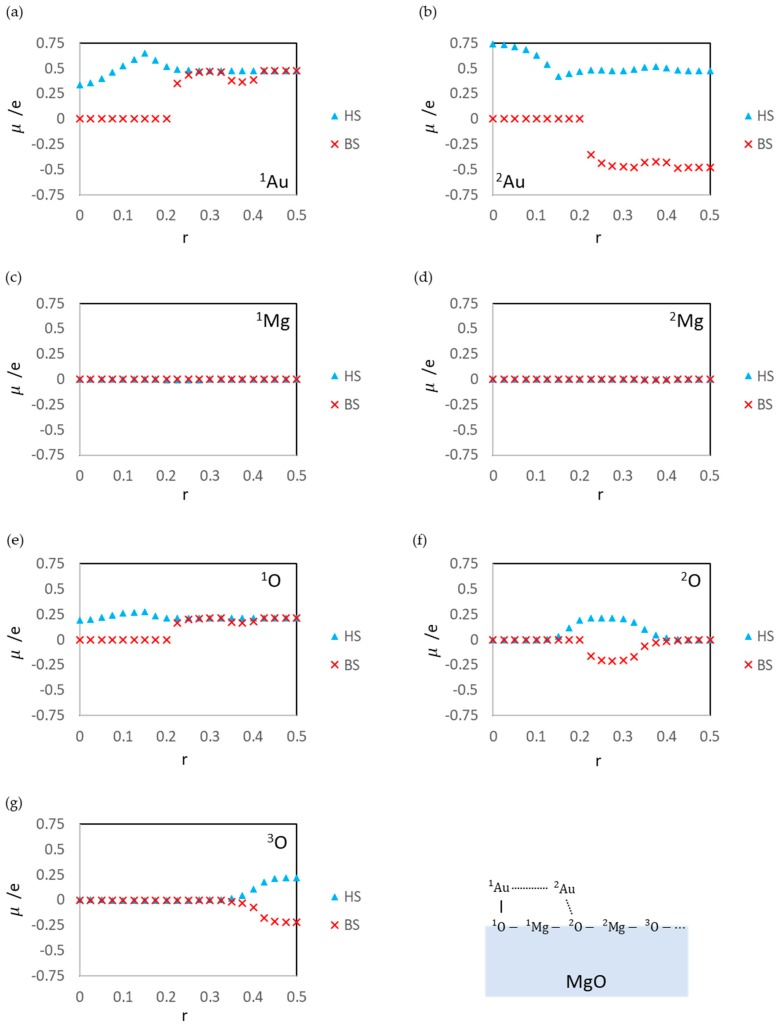
Local magnetic moments of ^1^Au (**a**), ^2^Au (**b**), ^1^Mg (**c**), ^2^Mg (**d**), ^1^O (**e**), ^2^O (**f**), and ^3^O (**g**) atoms in the optimized structures at each r value.

**Figure 11 molecules-24-00505-f011:**
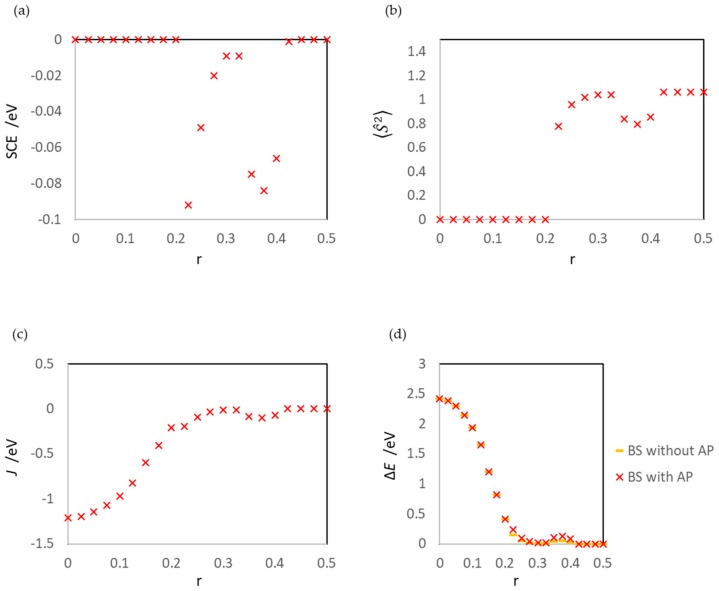
(**a**) Spin contamination error SCE, (**b**) expectation value of the square of the total spin operator $\langle {\hat{S}}^{2}\rangle$, (**c**) effective exchange integral *J* (Equation (5)), and (**d**) energy difference between HS and BS states ΔE (Equation (13)). The *J* values for r = 0.000–0.225 are estimated by using LS values; strictly, these values are not effective exchange integrals.

**Table 1 molecules-24-00505-t001:** Activation energies of reaction (1).

State	Aggregation of Au Atoms	Au_2_ Dissociation
LS	0.17 eV (0.53 eV ^1^)	2.47 eV
HS	0.45 eV	0.00 eV (2.39 eV ^2^)
BS	0.40 eV	2.34 eV
AP	0.32 eV	2.26 eV

^1^ Estimated using the BS value as baseline; ^2^ Estimated using the LS value as baseline.

**Table 2 molecules-24-00505-t002:** The values of SCE, $\langle {\hat{S}}^{2}\rangle$, and *J* of BS states for reactions (1) (surface reaction) and (2) (gas-phase reaction).

r	SCE/eV		$\langle {\hat{S}}^{2}\rangle$		*J*/eV		*d*(^1^Au-^2^Au)/Å
	Surface	Gas	Surface	Gas	Surface	Gas	
0.000	0.000	0.000	0.0000	0.0000	−1.208 ^1^	−0.894 ^1^	2.52917
0.025	0.000	0.000	0.0000	0.0000	−1.193 ^1^	−0.894 ^1^	2.53112
0.050	0.000	0.000	0.0000	0.0000	−1.148 ^1^	−0.893 ^1^	2.54011
0.075	0.000	0.000	0.0000	0.0000	−1.073 ^1^	−0.891 ^1^	2.55653
0.100	0.000	0.000	0.0000	0.0000	−0.966 ^1^	−0.891 ^1^	2.57930
0.125	0.000	0.000	0.0000	0.0000	−0.823 ^1^	−0.893 ^1^	2.61054
0.150	0.000	0.000	0.0000	0.0000	−0.600 ^1^	−0.880 ^1^	2.66684
0.175	0.000	0.000	0.0000	0.0000	−0.407 ^1^	−0.626 ^1^	2.99674
0.200	0.000	0.000	0.0000	0.0001	−0.207	−0.399 ^1^	3.40169
0.225	−0.092	0.000	0.7747	0.0001	−0.195	−0.207	3.82430
0.250	−0.049	−0.095	0.9567	0.7240	−0.093	−0.083	4.24920
0.275	−0.020	−0.054	1.0168	0.9215	−0.037	−0.031	4.67412
0.300	−0.009	−0.023	1.0394	0.9915	−0.017	−0.012	5.09924
0.325	−0.009	−0.009	1.0394	1.0178	−0.015	−0.004	5.52656
0.350	−0.075	−0.003	0.8399	1.0283	−0.088	−0.001	5.96645
0.375	−0.084	−0.001	0.7947	1.0325	−0.101	0.000	6.39464
0.400	−0.066	0.000	0.8554	1.0344	−0.075	0.000	6.81369
0.425	−0.001	0.000	1.0616	1.0353	−0.001	0.000	7.22546
0.450	0.000	0.000	1.0619	1.0358	−0.001	0.000	7.64864
0.475	0.000	0.000	1.0605	1.0360	0.000	0.000	8.07348
0.500	0.000	0.000	1.0599	1.0360	0.000	0.000	8.49840

^1^ Estimated by using the LS value.

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
