# Peer review of "Extent of Spin Contamination Errors in DFT/Plane-wave Calculation of Surfaces: A Case of Au Atom Aggregation on a MgO Surface"

_molecules, 2019, doi:10.3390/molecules24030505_

Round 1

Reviewer 1 Report

The authors study the effect of spin contamination error and static correlation on the aggregation of Au atoms on a MgO surface by restricted and unrestricted density functional theory calculations with a plane-wave basis and the approximate spin projection method. The authors find that the open-shell structure and the correction of the spin contamination error are important factors in calculating the small-cluster aggregations and molecule dimerization on surfaces.

The data is presented clearly in the manuscript and the discussion is detailed.

The manuscript is suggested to publish as it is.

Author Response

Thank you for your kind comments to our manuscript. We minor-revised the manuscript according to the other reviewer.

Reviewer 2 Report

In the manuscript, the authors implement unrestricted/restricted DFT with plane-wave basis and AP method to calculate the dissociation of Au2 dimer (aggregation of Au atoms) on the MgO surface. Their results show that the spin contamination error (SCE) affect the calculation of reaction in surface in the region 0.225<r<0.450, which is wider than in gas phase.  But the magnitude of the SCE is smaller in surface reaction. The SCE can be corrected by the AP method. This work is based on their previous publication (Chem. Phys. Lett. 2018,701, 103) and is more extensive and systematic. I recommend publication after minor revision to address the comments/suggestions below:

1. The title is confusing. The “spin contamination error” is an artificial error which affect the accuracy of the “calculation”, not the “aggregation of Au atoms”. How the aggregation reaction occurs has no connection with how the calculation is done. Thus, the title needs to be clearer.

2. Line 153, how are the lengths of supercell determined? Will different lengths lead to different Au atom energies?

3. In section 3, the authors arrange the subsections in the order of Au2 dissociation with the r value increasing (except 3.2). However, the authors concluded the “investigated reaction” in line 490 to 498 in the Au atoms aggregation direction. It would be better to rearrange section 3 to follow the same direction (Au atoms aggregation with r decreasing), which is also consistent with the title.

4. Line 469, “sates” should be “states”.

Author Response

Thank you for your comments to our manuscript. We revised the manuscript according to your comments. The detailes are in the attached file.
